

# Effects of different training modalities on phosphate homeostasis and local vitamin D metabolism in rat bone

Joost Buskermolen[1], Karen van der Meijden[2], Regula Furrer[3], Dirk-Jan Mons[1], Huib W. van Essen[1], Annemieke C. Heijboer[1], Paul Lips[2], Richard T. Jaspers[3] and Nathalie Bravenboer[1]

[1] Department of Clinical Chemistry, VU University Medical Center, Amsterdam, The Netherlands
[2] Department of Internal Medicine/Endocrinology, VU University Medical Center, Amsterdam, The Netherlands
[3] Laboratory for Myology, VU University Amsterdam, Amsterdam, The Netherlands

## ABSTRACT

**Objectives.** Mechanical loading may be an important factor in the regulation of bone derived hormones involved in phosphate homeostasis. This study investigated the effects of peak power and endurance training on expression levels of fibroblast growth factor 23 (FGF23) and $1\alpha$-hydroxylase (CYP27b1) in bone.

**Methods.** Thirty-eight rats were assigned to six weeks of training in four groups: peak power (PT), endurance (ET), PT followed by ET (PET) or no training (control). In cortical bone, FGF23 was quantified using immunohistochemistry. mRNA expression levels of proteins involved in phosphate and vitamin D homeostasis were quantified in cortical bone and kidney. C-terminal FGF23, 25-hydroxyvitamin D3, parathyroid hormone (PTH), calcium and phosphate concentrations were measured in plasma or serum.

**Results.** Neither FGF23 mRNA and protein expression levels in cortical bone nor FGF23 plasma concentrations differed between the groups. In cortical bone, mRNA expression levels of sclerostin (SOST), dental matrix protein 1 (DMP1), phosphate-regulating gene with homologies to endopeptidases on the X chromosome (PHEX) and matrix extracellular phosphoglycoprotein (MEPE) were lower after PT compared to ET and PET. Expression levels of CYP27b1 and vitamin D receptor (VDR) in tibial bone were decreased after PT compared to ET. In kidney, no differences between groups were observed for mRNA expression levels of CYP27b1, 24-hydroxylase (CYP24), VDR, NaPi-IIa cotransporter (NPT2a) and NaPi-IIc cotransporter (NPT2c). Serum PTH concentrations were higher after PT compared to controls.

**Conclusion.** After six weeks, none of the training modalities induced changes in FGF23 expression levels. However, PT might have caused changes in local phosphate regulation within bone compared to ET and PET. CYP27b1 and VDR expression in bone was reduced after PT compared to ET, suggesting high intensity peak power training in this rat model is associated with decreased vitamin D signalling in bone.

Corresponding author
Nathalie Bravenboer,
n.bravenboer@vumc.nl

## INTRODUCTION

Mechanical forces associated with physical exercise cause deformation of bone tissue. These deformations activate an acute response in osteocytes, in particular production of cytokines and signalling molecules such as nitric oxide (NO), bone morphogenetic proteins (BMPs), Wnts, and prostaglandin E2 (PGE2) (*Klein-Nulend, Bacabac & Bakker, 2012*). These factors affect osteoblast and osteoclast recruitment, differentiation and activation locally at the sites that are subjected to high strains. Since whole bone response to mechanical loading depends on the strain rate and magnitude (*Rubin & Lanyon, 1985*; *Turner, Owan & Takano, 1995*), peak power training with high peak loads with a high frequency is generally considered to result in a higher bone response than endurance training. Moreover, it was reported that a combination of training types even better maintained bone mineral density in postmenopausal women (*Martyn-St James & Carroll, 2009*).

Recently, it was proposed that mechanical loading affects gene expression and local protein activity of fibroblast growth factor 23 (FGF23) (*Sapir-Koren & Livshits, 2014*). FGF23 mRNA expression in mouse bone has been shown to increase after 6 days of endurance training (*Gardinier et al., 2016*). Moreover, it was suggested FGF23 might be a 'molecular mediator of the whole-body effects of exercise originating from bone' (*Qi, Liu & Lu, 2016*). Indeed, serum FGF23 increased after endurance exercise in rat and human (*Li et al., 2016*; *Lombardi et al., 2014*), indicating a potential systemic effect of local mechano-response in bone; FGF23 is produced exclusively in bone but acts on the kidneys, where it decreases the reabsorption and increases excretion of phosphate and also suppress $1\alpha$-hydroxylase (CYP27b1), reducing its ability to activate vitamin D and subsequently impairing calcium absorption (*Bonewald & Wacker, 2013*). To our knowledge, the direct effect of different training types on FGF23 expression within bone has not been reported yet.

A known stimulator of FGF23 production is systemic 1,25-dihydroxyvitamin D3 (1,25(OH)2D3) (*Saji et al., 2010*) or *in vitro* locally produced 1,25(OH)2D3 (*Tang et al., 2010*). Inhibitors of FGF23 production are mineralization-regulating proteins such as phosphate-regulating gene with homologies to endopeptidases on the X chromosome (PHEX) and dental matrix protein 1 (DMP1) (*Martin et al., 2011*), possibly through alterations in bone mineralization. Mechanical loading could either affect FGF23 gene expression directly, or indirectly by alterations in the expression of FGF23-regulating proteins known to be stimulated by mechanical loading, such as DMP1 (*Gluhak-Heinrich et al., 2003*; *Yang et al., 2005*) and matrix extracellular phosphoglycoprotein (MEPE) (*Reijnders et al., 2013*) as well as presumably locally converted 1,25(OH)2D3.

The active metabolite 1,25(OH)2D3 is hydroxylated from 25-hydroxyvitamin D3 (25(OH)D3) by $1\alpha$-hydroxylase (*Lips, 2006*), which is, among other tissues, expressed by human bone cells (*Van Driel et al., 2006*; *Van der Meijden et al., 2014*). The regulation of activity of $1\alpha$-hydroxylase within bone is yet to be determined. *Van Driel et al. (2006)* demonstrated that in cultured bone cells 25(OH)D3, PTH and calcium did not affect CYP27b1 mRNA expression and activity. In contrast, treatment of primary human osteoblasts with high doses of calcium did increase CYP27b1 mRNA expression (*Van*

*der Meijden et al., 2016c*). Pulsatile fluid flow, an *in vitro* model for mechanical loading, increased CYP27b1 expression in primary human osteoblasts (*Van der Meijden et al., 2016a*), suggesting that 1α-hydroxylase within bone is at least partly regulated by mechanical loading. Among healthy, young women, serum 25(OH)D3 did not differ between different training groups (*Lester et al., 2009*). However, among rats an increase in serum 1,25(OH)2D3 after endurance training compared to controls was observed (*Iwamoto et al., 2004*). Whether vitamin D signalling within bone is affected by mechanical loading *in vivo* remains to be elucidated.

Osteocytes may respond to mechanical loading by altering the expression of FGF23 and CYP27b1. As the response of osteocytes to mechanical loading is determined by the type of loading, FGF23 production and 1α-hydroxylation may be differentially changed according to the type of loading. We hypothesize that mechanical loading induced by physical exercise influences FGF23 production and 1α-hydroxylation in bone, with the greatest response after a combination of both peak power training and endurance training. Therefore, the aim of this study is to investigate how different kinds of training modalities affect the production of phosphate regulating proteins and 1α-hydroxylation by rat bone tissue *in vivo*.

## MATERIALS AND METHODS

### Experimental design and training protocol

The animal experiment was approved by the Animal Experiment Committee of the VU University Amsterdam with permit number FBW 10-03, and described previously (*Furrer et al., 2013*). Briefly, 38 female Wistar rats at the age of 13 weeks were assigned to four groups: control ($n = 8$), peak power training (PT, $n = 10$), endurance training (ET, $n = 10$) and a combined training (PET, $n = 10$). The rats were trained on a treadmill for 6 weeks, 5 days a week for 1 session a day (ET and PT group). For the peak power training, rats performed 10 sprints of 15 s in gallop at a maximal attainable velocity on a progressively increasing slope starting at 10% reaching up to 40% by the end. Endurance training consisted of treadmill running for 10 min at a speed of 16 m/min without a slope which was gradually increased up to 45 min of length with a speed of 26 m/min (trotting) on a 10% slope. The rats following a combination of training types (PET) conducted 2 training sessions a day: peak power training in the morning and endurance training 8 h later. Previously, an 8 h rest period was shown sufficient to restore full mechanosensitivity (*Robling, Burr & Turner, 2001*). All rats were sacrificed 22 h after the last training by cardiac injection with Euthasol 20% (AST Farma B.V., Oude water, The Netherlands). The animals were group housed, rodent diet (Teklad Global 16% Protein Rodent Diet, Madison, WI, USA) containing 1% calcium, 0.7% phosphorus and 1.5 IU/g vitamin D3 was provided *ad libitum* and the rats were kept on a reversed 12 h light/dark cycle to provide the training during their active period of the day.

### Immunohistochemistry

Immunohistochemistry was performed on the right tibial shaft to analyse protein expression of FGF23. Non-decalcified tibiae were fixed in 4% phosphate buffered paraformaldehyde

and after dehydration in increasing alcohol series, embedded in 80% methylmethacrylate (MMA) (BDH Chemicals, Poole, England) with 20% dibuthylphtalate (Merck, Darmstadt, Germany). Longitudinal five-micrometer-thick tissue sections were cut using a Polycut 2500 S microtome (Reichert-Jung, Nussloch, Germany). To remove MMA, tissue sections were incubated in 50% xylene and 50% chloroform. Subsequently, sections were rehydrated in a series of decreasing alcohol concentrations. Decalcification was done with 1% acidic acid, followed by quenching of endogenous peroxidases with 3% H2O2 in 40% methanol/PBS. Tissue sections were incubated with 10% goat serum to prevent unspecific binding of the secondary antibody. Subsequently, sections were incubated overnight at 4 °C with polyclonal 1:200 rabbit anti-FGF23 (AB_2104625, Santa Cruz Biotechnology Sc-50291, CA, USA). The next day, tissue sections were incubated with polyclonal 1:100 biotinylated goat-anti-rabbit (AB_2313609, Dako E0432, Heverlee, Belgium) for 1 h and with 1:200 horseradish peroxidase labelled streptavidin (Invitrogen, Life Technologies, Bleiswijk, The Netherlands) for 1 h. Signal enhancement was established by treatment with tyramide (Invitrogen, Life Technologies, Bleiswijk, The Netherlands) for 10 min followed by a second 1 h incubation with 1:200 horseradish peroxidase labelled streptavidin. Chromogenesis was performed by treatment of the sections with AEC reagent (Invitrogen, Life Technologies, Bleiswijk, The Netherlands) for 6 min and by counterstaining with haematoxylin. Finally, the sections were mounted with ClearMount Mounting Solution (Invitrogen, Life Technologies, Bleiswijk, The Netherlands) and covered with a coverslip.

For each rat, two or three longitudinal tissue sections (depending on the quality of the tissue section), separated by 150 µm, were analysed using an Olympus BX51 Microscope at 200× magnification. The entire length of the tibial shaft was divided in alternating longitudinal regions of interest of 50 µm long, in which all osteocytes across the entire thickness of the cortex were counted manually. Per tissue section, up to 400 osteocytes were counted. The amount of positively stained osteocytes was compared to the total number of osteocytes and expressed as fraction positively stained osteocytes compared to the total number of osteocytes.

## Quantitative polymerase chain reaction (qPCR)
### Tissue preparation and RNA-extraction

Left tibiae and kidneys were snap frozen and stored at −80 °C until further analysis. One week before tibial RNA-isolation and 40 h prior to kidney RNA-isolation, tibiae and kidneys were stored in RNA*later*®-ICE (Ambion, Life Technologies, Bleiswijk, The Netherlands). The kidneys were homogenised using a Dounce Homogeniser (Sigma-Aldrich, Zwijndrecht, The Netherlands). Subsequent processing occurred according to protocol from the column-based 'FavorPrep Tissue Total RNA Purification Maxi Kit' (Favorgen Biotech corp., Huissen, The Netherlands). For RNA isolation of the tibiae, proximal and distal ends of the tibiae were cut off. Bone marrow was removed by flushing the diaphysis with ice cold RNAse free water. Diaphyseal cortical bone was pulverised with a freezer-mill (SPEX 6750, Glen Creston, Stanmore, England) and incubated in Trizol (Invitrogen, Life Technologies, Bleiswijk, The Netherlands) for 1 h at 37 °C. After the first Trizol-extraction, a chloroform-isomyl alcohol extraction was performed, followed by a

**Table 1  Details of primers used for quantitative PCR analysis.**

| Target gene | PCR primer sequences 5′−>3′ | |
| --- | --- | --- |
| | Forward | Reverse |
| CYP24 | GCT-GAT-GAC-AGA-CGG-TGA-GA | TGT-CGT-GCT-GTT-TCT-TCA-GG |
| CYP27b1 | CCC-GAC-ACA-GAA-ACC-TTC-AT | GGC-AAA-CAT-CTG-ATC-CCA-GT |
| DMP1 | GCG-ACT-CCA-CAG-AGG-ATT-TC | GTC-CCT-CTG-GGC-TAT-CTT-CC |
| FGF23 | GAT-GCT-GGC-TCC-GTA-GTG-AT | CGT-CGT-AGC-CGT-TCT-CTA-GC |
| HPRT | GTG-TCA-TCA-GCG-AAA-GTG-GA | TAC-TGG-CCA-CAT-CAA-CAG-GA |
| MEPE | AAG-ACA-AGC-CAC-CCT-ACA-CG | CCC-ACT-GGA-TGA-TGA-CTC-ACT |
| NPT2a | AGT-GGC-CAA-TGT-CAT-CCA-GA | AGT-GAT-GGC-TGA-GGT-GAA-CA |
| NPT2c | GGT-CAC-CGT-CCT-TGT-ACA-GA | GAC-GCC-CAT-GAT-GAT-AGG-GA |
| PBGD | ATG-TCC-GGT-AAC-GGC-GGC | CAA-GGT-TTT-CAG-CAT-CGC-TAC-CA |
| PHEX | CAG-GCA-TCA-CAT-TCA-CCA-AC | GGA-GGA-CTG-TGA-GCA-CCA-AT |
| SOST | CAG-CTC-TCA-CTA-GCC-CCT-TG | GGG-ATG-ATT-TCT-GTG-GCA-TC |
| VDR | ACA-GTC-TGA-GGC-CCA-AGC-TA | TCC-CTG-AAG-TCA-GCG-TAG-GT |

**Notes.**

CYP24, 24-hydroxylase; CYP27b1, 1 $\alpha$-hydroxylase; DMP1, dental matrix protein 1; FGF23, fibroblast growth factor 23; HPRT, hypoxanthine phosphoribosyltransferase; MEPE, matrix extracellular phosphoglycoprotein; NPT2a, NaPi-IIa co-transporter; NPT2c, NaPi-IIc cotransporter; PBGD, porphobilinogendeaminase; PHEX, phosphate-regulating gene with homologies to endopeptidases on the X chromosome; SOST, sclerostin; VDR, vitamin D receptor.

second Trizol-extraction. Both kidney and tibial RNA were treated with DNAse (Promega, Leiden, The Netherlands) to remove DNA-contamination.

### Reversed transcription

After RNA-isolation, 100 ng of total RNA was reverse-transcribed using 10 ng/µl random primers (Roche, Almere, The Netherlands) and 5 U/µl M-MLV Reverse Transcriptase in a mixture with 5 mM MgCl$_2$, 1x RT-buffer, 1 mM dNTPs each, 1M betaine and 0.40 U/µl RNAsin (Promega, Leiden, The Netherlands). A total volume of 20 µl was incubated for 10 min at 25 °C, 1 h at 37 °C and 5 min at 95 °C.

### Quantitative PCR

A total volume of 25 µl containing 3 µl of cDNA, 1000 nM primers and 12,5 µl SYBR Green Supermix (Bio-Rad, Veenendaal, The Netherlands) was amplified in the iCycler system (Bio-Rad, Veenendaal, The Netherlands) using the primers as described in Table 1. PCR consisted of an initial denaturation step for 3 min at 95 °C, followed by 40 amplification cycles (15 s at 95 °C, 1 min at 60 °C). Subsequently, a melting curve was run from 50 °C to 95 °C to check the specificity of the reactions.

The following mRNA levels were assessed in the tibia: 1$\alpha$-hydroxylase (CYP27b1), 24-hydroxylase (CYP24), vitamin D receptor (VDR), fibroblast growth factor 23 (FGF23,) dental matrix protein 1 (DMP1), phosphate-regulating gene with homologies to endopeptidases on the X chromosome (PHEX), matrix extracellular phosphoglycoprotein (MEPE) and sclerostin (SOST).

The following mRNA levels were assessed in the kidney: CYP24, CYP27b1, VDR, NaPi-IIa cotransporter (NPT2a or SLC34a1) and NaPi-IIc cotransporter (NPT2c or SLC34a3).

Levels of tibial and kidney mRNA were expressed relative to the average of reference genes hypoxanthine phosphoribosyltransferase (HPRT) and porphobilinogendeaminase (PBGD) using the $2^{-\Delta CT}$ method. All samples were assessed in duplicate or triplicate on a single 96 well-plate per gene.

### Serum biochemical analysis

Blood was obtained by puncture of the vena cava during general anaesthesia prior to sacrifice. Rat C-terminal FGF23 Elisa Kit (Immutopics, San Clemente, CA, USA) was used to measure FGF23 in EDTA plasma. Serum 25(OH)D3 was detected using liquid chromatography-tandem mass spectrometry (LC-MS/MS) according to standardised methods (*Van der Meijden et al., 2016b*). Serum PTH was measured using the Rat Intact PTH Elisa Kit (Immutopics, San Clemente, CA, USA). Serum calcium and phosphate concentrations were measured using an Elecsys platform (Roche Diagnostics, Mannheim, Germany).

### Statistical analysis

GraphPad Prism 6.0 (GraphPad Software Inc., La Jolla, CA, USA) was used for data analysis. Differences between the four groups were tested with a Kruskal-Wallis Test, followed by a Dunn's post-hoc test. A value of $p < 0.05$ was considered to be significant. Results are reported as means per group. Within figures, each dot represents the mean value of a single rat, whereas the bars represent the means per group.

## RESULTS

### Animal experiment

Of the thirty rats that performed training, three rats (two of the PT group and one of the ET group) were excluded for not fulfilling the required training protocol. This resulted in a total sample size for controls $n = 8$, ET $n = 9$, PT $n = 8$ and PET $n = 10$. The final body weight of the included rats ($n = 35$) did not differ among the groups. Tibial RNA-isolation for one control rat failed. We could not collect plasma samples for one control, three ET-rats, two PT-rats and one PET-rat.

### FGF23 protein expression in tibiae

As shown in Fig. 1, immunohistochemical analysis of FGF23 protein expression by cortical osteocytes in the tibiae did not reveal any differences in the number of stained cells between the four groups. FGF23 staining in trabecular bone could unfortunately not be quantified, due to low contrast between positive osteocytes and aspecific staining of the marrow.

### mRNA expression in tibiae

Figure 2A shows that FGF23 mRNA expression levels did not differ amongst the groups. mRNA levels of osteocyte maturation markers SOST, DMP1, PHEX and MEPE were significantly lower after PT compared to those after ET and PET, as illustrated by Figs. 2B–2E, respectively. Furthermore, Figs. 2F and 2G show that CYP27b1 and VDR mRNA expression levels were lower after PT compared to those after ET. Expression of CYP24 mRNA was below detection limit.

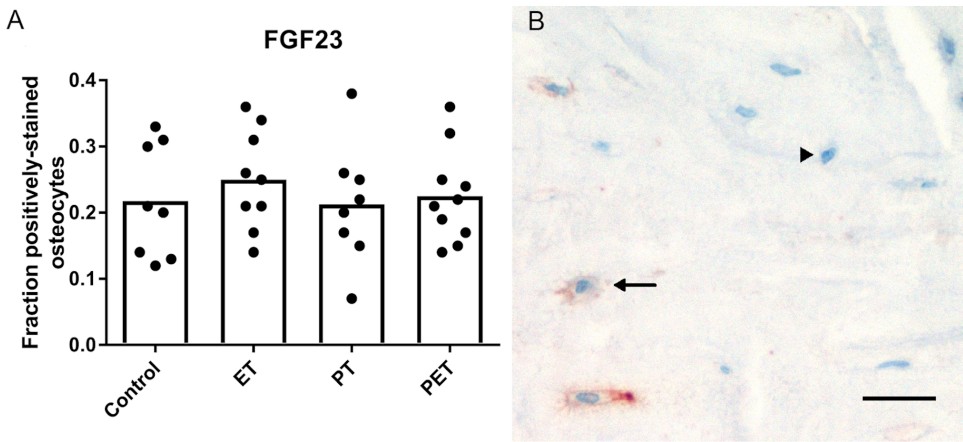

**Figure 1** **Effects of different training modalities on the expression of FGF23 in rat tibiae as analysed by immunohistochemistry.** (A) Fraction positively stained cortical osteocytes compared to total amount of cortical osteocytes. (B) Representative image of FGF23 positively stained cortical osteocytes after PET. Arrows show positively stained osteocytes, whereas arrowheads point out FGF23 negative osteocytes. Bar represents 25 μm (ET, endurance training; PT, peak training; PET, combined peak- and endurance training).

## mRNA expression in kidneys

Assuming that FGF23 production would be altered by physical exercise, we investigated mRNA expression levels of FGF23 responsive genes in the kidney to assess whether this would have systemic influences. Figure 3 reveals that there were no significant differences in CYP27b1, CYP24, VDR, NPT2a and NPT2c mRNA expression levels. Therefore, in kidney tissue physical exercise did not induce any changes in mRNA expression levels of FGF23-responsive genes and VDR.

## Serum biochemical analysis

Figure 4 shows there were no significant differences in plasma c-term FGF23 concentration between groups. However, we did see an increase in serum PTH concentration in the PT-group compared to the controls. Serum 25(OH)D3, calcium and phosphate levels were not significantly different between groups.

## DISCUSSION

This study aimed to investigate whether different kinds of training modalities affect phosphate homeostasis and 1α-hydroxylation in rat bone *in vivo*.

In contrast to our hypothesis, we observed no differences between groups in FGF23 expression in both bone and serum. In line with our results, FGF23 serum levels were not affected by either submaximal exercise or high intensity exercise with a bicycle ergometer among young men (*Emrich et al., 2018*). Yet, *Lombardi et al. (2014)* described increased FGF23 serum concentrations among participants of the multiple-stage bicycle race Giro d'Italia. However, confounding factors in that study could be the high dietary intake as well as weight loss, suggesting the observed changes might be an effect of altered metabolic

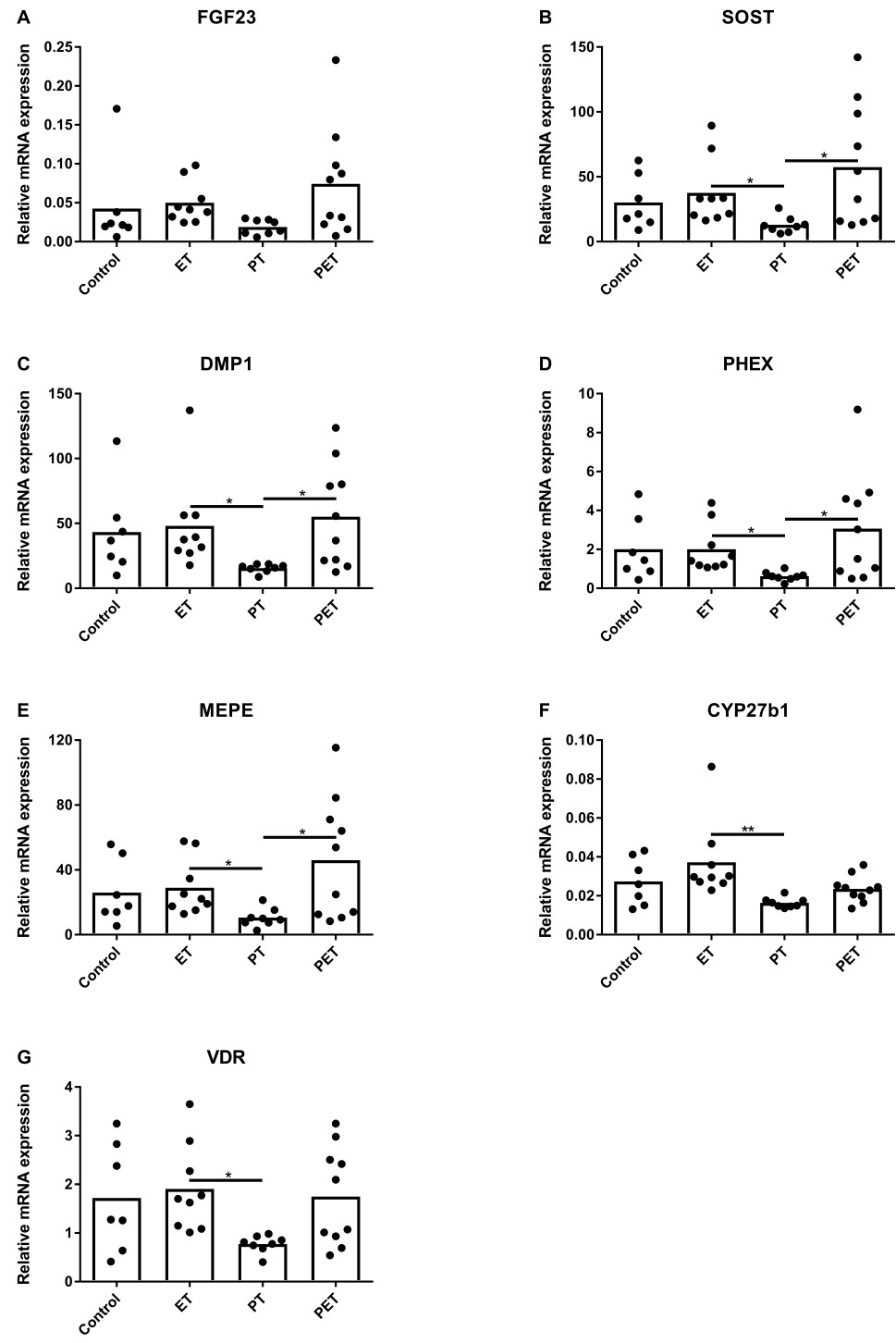

**Figure 2 Effects of different training modalities on mRNA expression in rat tibiae as analysed by qPCR.** Gene expression levels of (A) FGF23, (B) SOST, (C) DMP1, (D) PHEX, (E) MEPE, (F) CYP27b1 and (G) VDR were measured. Results were normalized for reference genes HPRT and PBGD. Significant differences are indicated as $*p < 0.05$ and $**p < 0.01$ (ET, endurance training; PT, peak training; PET, combined peak- and endurance training).

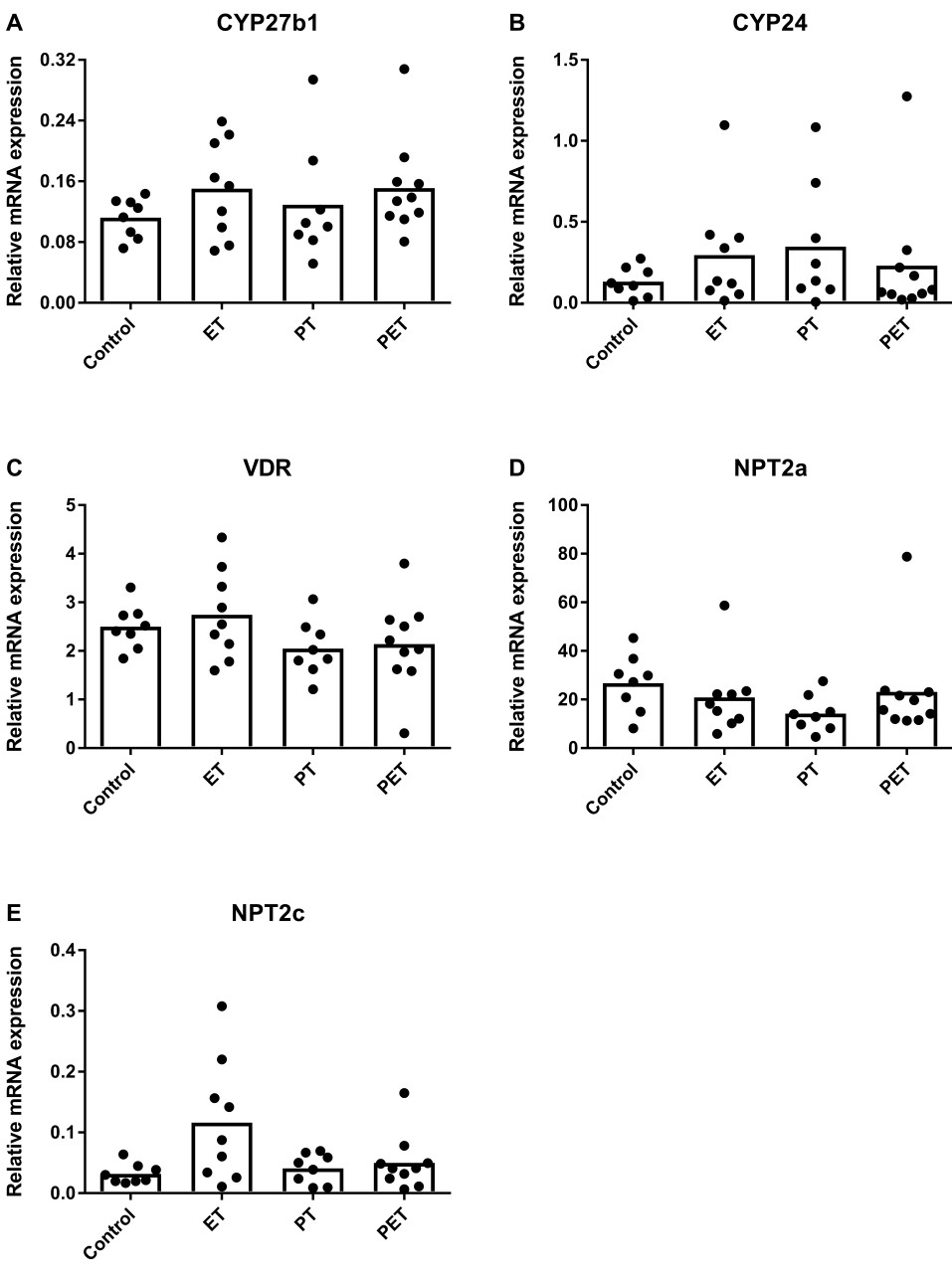

**Figure 3** **Effects of different training modalities on mRNA expression in rat kidneys as analysed by qPCR.** Gene expression levels of (A) CYP27b1, (B) CYP24, (C) VDR, (D) NPT2a and (E) NPT2c were measured. Results were normalized for reference genes HPRT and PBGD. Significant difference is indicated as $^*p < 0.05$ (ET, endurance training; PT, peak training; PET, combined peak- and endurance training).

state rather than mechanical loading of bone cells. Moreover, these cyclists showed signs of induced bone resorption (*Lombardi et al., 2012*), possibly as a consequence of the heavy metabolic stress in the absence of mechanical loading, whereas we assume a situation of bone formation after mechanical loading in our model. Our findings also stand in contrast

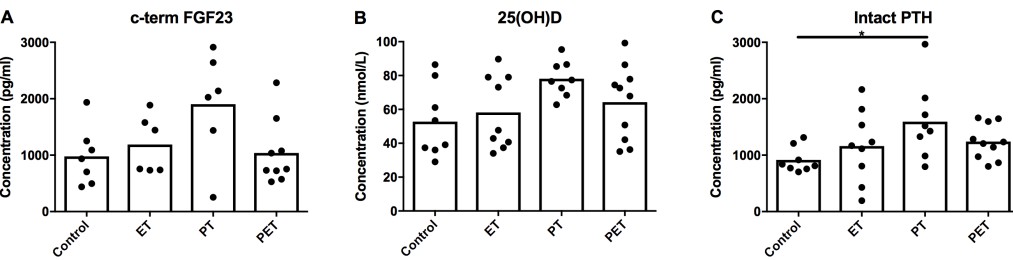

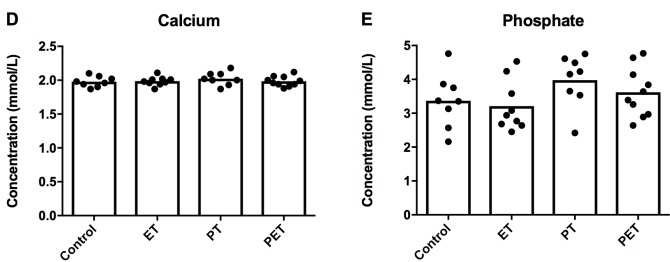

**Figure 4** **Effects of different training modalities on serum concentrations of factors involved in phosphate homeostasis.** Serum concentrations of (A) c-terminal FGF23, (B) 25(OH)D3, (C) parathyroid hormone (PTH), (D) calcium and (E) phosphate were assessed. Significant difference is indicated as $*p < 0.05$.

with those of *Li et al. (2016)*, who reported increased serum FGF23 after acute exercise, exhaustive exercise, and chronic exercise among mice. However, these mice trained for only one week, whereas the rats in our model performed exercise for up to 6 weeks. As Li et al. also described that FGF23 promotes exercise endurance, it is possible the rats in our study at first responded to training with FGF23-upregulation to adapt to the exercise, but that after 6 weeks the rats were adapted and thus no longer showed increased FGF23 at the time of measurement.

Quantification of FGF23 substrates, such as genes involved in vitamin D metabolism and sodium-phosphate co-transporters in kidney tissue, ruled out that post-translational modification differed amongst groups. As FGF23 is considered the major phosphate regulating hormone (*Bonewald & Wacker, 2013*), it is not surprising that the lack of change in FGF23 or FGF23 substrates was paralleled with unchanged phosphate concentrations in serum after 6 weeks of training. Therefore, these results indicate that FGF23 production in bone and systemic phosphate homeostasis were not affected after 6 weeks of treadmill running.

Surprisingly, PT caused a decrease in both CYP27b1 and VDR mRNA levels in tibial bone compared to mRNA levels after ET, suggesting decreased vitamin D signalling in bone in the PT group. We had expected to observe an increase in local vitamin D signalling after PT compared to ET. Possibly, PT might have been too strenuous, causing fatigue damage of the bone leading to apoptosis of osteocytes (*Jilka, Noble & Weinstein, 2013*). This process

could also have accounted for the observed decreased vitamin D signalling after PT. In line with other studies (*Lombardi et al., 2014*; *Lester et al., 2009*), no differences between groups were observed for serum 25(OH)D concentrations, suggesting the observed decreased vitamin D signalling is rather a local process within bone than a systemic effect of exercise.

We hypothesized that alterations in FGF23 expression and $1\alpha$-hydroxylation in bone tissue in response to mechanical loading are related; both FGF23 expression and $1\alpha$-hydroxylation are likely to be altered by mechanical loading and both processes may be involved in regulation of each other. As we did not observe differences in FGF23 production but did observe decreased vitamin D signalling in bone tissue, we cannot confirm this hypothesis. Previous studies show contradicting results regarding this issue. In cultured rat osteoblasts FGF23 mRNA expression levels appeared to be enhanced by "autocrine/paracrine action of osteoblast-derived $1\alpha$, 25(OH)2D" (*Tang et al., 2010*). In contrast, incubating cultured primary human osteoblasts with FGF23 did not cause changes in mRNA expression of CYP27b1 and VDR (*Van der Meijden et al., 2016c*). Moreover, in human bone samples an association between FGF23 and CYP27b1, CYP24 or VDR mRNA expression could not be shown (*Ormsby et al., 2013*). Possibly, in our rat model an acute change in FGF23 expression and $1\alpha$-hydroxylation in bone tissue may have occurred, but a new bone balance between the two was established at the time of investigation.

Furthermore, PTH levels were higher after PT compared to controls 22 h after the last training. It is likely that the observed PTH concentrations after PT reflect an increased basal level of PTH in this group after six weeks of training. In general, a transient increase of PTH is observed during and directly after exercise, depending on the type of exercise (*Maimoun & Sultan, 2009*), and most studies report normalization of PTH serum concentrations shortly after exercise (*Maimoun & Sultan, 2009*; *Scott et al., 2011*; *Gardinier, Mohamed & Kohn, 2015*), but some studies observed a prolonged increase in PTH 24 h after a single exercise (*Brahm, Piehl-Aulin & Ljunghall, 1997*; *Bouassida et al., 2003*; *Thorsen et al., 1997*). Low basal PTH concentrations are associated with higher physical fitness (*Brahm, Piehl-Aulin & Ljunghall, 1997*).

Another interesting observation is that PT compared to ET and PET showed lower expression of osteocyte maturation markers SOST, DMP1, PHEX and MEPE. Therefore, it seems that PT caused a suppression of mature osteocyte function, which was reversed by additional endurance training. In general, DMP1 and MEPE are known to increase after mechanical loading (*Gluhak-Heinrich et al., 2003*; *Yang et al., 2005*; *Reijnders et al., 2013*), whereas SOST is downregulated by mechanical loading (*Robling et al., 2008*; *Gardinier et al., 2016*). DMP1, MEPE and PHEX are not only markers of osteocyte maturation, but are functionally involved in phosphate regulation and bone mineralization (*Rowe, 2012*). DMP1 and PHEX are known to be involved in inhibition of FGF23 production (*Martin et al., 2011*). Therefore, it was expected that reduction of these factors would lead to an increase in FGF23 production in the PT group in our rat model. However, after 6 weeks of peak training FGF23 production and function in systemic phosphate homeostasis appeared unchanged, which does not exclude that local changes in phosphate regulation in bone tissue may have occurred after PT. Moreover, it has been suggested that DMP1 and MEPE cause local changes in stiffness and mineralization of canaliculi and lacunae, thereby

altering the osteocytic response to mechanical loading (*Harris et al., 2007*). Possibly a new balance has been established between DMP1, MEPE and FGF23 after 6 weeks of peak training, in which FGF23 carries out its systemic functions whereas DMP1 and MEPE have a local function in lacunar remodelling.

Local mechano-response in bone has the potential to lead to systemic effects, because bone is an endocrine organ (*Han et al., 2018*), and changes in bone resorption and formation can lead to alterations in systemic calcium and phosphate. However, in comparison to the many local mechano-responsive changes in tibial bone in this rat model, the only observed systemic effect was an alteration of PTH concentrations after PT. PTH concentrations respond to many other stimuli than just mechanical loading, such as changes in the adrenergic system, pH and or lactic acid (*Bouassida et al., 2006*). As neither systemic FGF23, 25(OH)D3, calcium and phosphate, nor any of the genes measured in kidney tissue differed between groups, it seems that the effects of mechanical loading in this model did not lead to major systemic differences between groups.

Remarkably, we did not observe any statistically significant effect of PET compared to controls or ET. Therefore, it seems that the observed changes in gene and protein expression after PT have been reversed by additional ET. These results should be interpreted with caution; the rats performing PET had more training than the other groups. By performing both training programs, they underwent a higher amount of strain, which might be a confounding factor.

In this study organs were harvested 22 h after the last training session in order to study contractile muscle force characteristics *in situ* (*Furrer et al., 2013*). Therefore, acute effects of mechanical loading on mRNA concentrations might have already disappeared at the time of investigation. The observed differences after PT might indicate a possible new steady state.

Another limiting factor to our study is the lack of accurate FGF23-staining in trabecular bone. Trabecular bone is generally considered to be metabolic active and possible effects of exercise on FGF23 protein expression within bone marrow could unfortunately not be quantified.

## CONCLUSION

In conclusion, six weeks of physical exercise did not cause significant differences in phosphate homeostasis on a systemic level. However, mRNA expression levels of genes involved in local phosphate regulation and vitamin D signalling within bone tissue differed among groups, suggesting a local mechano-response rather than a systemic response in this rat model.

## ACKNOWLEDGEMENTS

We would like to thank Ina Kamphuis and Mirjam Bethlehem for their assistance in cutting the tissue sections. Also, we are thankful to the technicians of the Endocrine Laboratory of the VU University Medical Center for the biochemical analyses.

### Funding
The authors received no funding for this work.

### Competing Interests
The authors declare there are no competing interests.

### Author Contributions
- Joost Buskermolen performed the experiments, analyzed the data, prepared figures and/or tables, authored or reviewed drafts of the paper, approved the final draft.
- Karen van der Meijden performed the experiments, analyzed the data, authored or reviewed drafts of the paper, approved the final draft.
- Regula Furrer and Dirk-Jan Mons performed the experiments, approved the final draft.
- Huib W. van Essen performed the experiments, analyzed the data, contributed reagents/materials/analysis tools, prepared figures and/or tables, approved the final draft.
- Annemieke C. Heijboer contributed reagents/materials/analysis tools, approved the final draft.
- Paul Lips authored or reviewed drafts of the paper, approved the final draft.
- Richard T. Jaspers conceived and designed the experiments, performed the experiments, analyzed the data, contributed reagents/materials/analysis tools, authored or reviewed drafts of the paper, approved the final draft.
- Nathalie Bravenboer conceived and designed the experiments, performed the experiments, analyzed the data, contributed reagents/materials/analysis tools, prepared figures and/or tables, authored or reviewed drafts of the paper, approved the final draft.

### Animal Ethics
The following information was supplied relating to ethical approvals (i.e., approving body and any reference numbers):

The animal experiment was approved by the Animal Experiment Committee of the VU University Amsterdam with permit number FBW 10-03.

### Data Availability
The raw measurements are provided in the Supplementary File. All raw data are included.

### Supplemental Information
Supplemental information for this article can be found online at http://dx.doi.org/10.7717/peerj.6184#supplemental-information.

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
