# Peer review of "Effects of different training modalities on phosphate homeostasis and local vitamin D metabolism in rat bone"

_PeerJ, doi:10.7717/peerj.6184_

## Round 0.1 · original submission · Major Revisions

Dear Dr. Bravenboer,

Your manuscript entitled “Effects of different training modalities on phosphate homeostasis and local vitamin D metabolism in rat bone" which you submitted to PeerJ, has been reviewed by the editor and 2 external reviewers.

I regret to inform you that the reviewers have raised some significant concerns that need to be addressed before the manuscript can be considered further.

First, I think it is important to clarify that, according to PeerJ editorial criteria, a manuscript should be judged based on determination of scientific and methodological soundness, not on subjective determinations of “impact”, “novelty”, “degree of advance” or 'interest”. Negative / inconclusive results are also acceptable, as they are not a sign of poorly conducted research. I think you (both authors and reviewers) can agree that the relevance of the data depends on the questions requiring answers and correctness of the postulates ,not on the outcome of the response. I take this opportunity to point out the link to PeerJ's editorial criteria (https://peerj.com/about/editorial-criteria/). Both authors and reviewers should not depart from these criteria.

For these reasons, I would be willing to reconsider if you wish to undertake major revisions and resubmit. However, please note that resubmitting your manuscript does not guarantee eventual acceptance. Since the requested changes are major, the revised manuscript will undergo a second round of review by the same reviewers. If you decide to resubmit the revised version, please summarize all the improvements made in the new version and give answers to all critical points raised in the reviewers’ report in an accompanying letter.

I suggest to pay particular attention to the following reviewers’ criticisms/suggestions:

1) measure serum levels of vitamin D (Reviewrs 1 an 2) and urinary levels of phosphate (Reviewer 1);
2) provide information on the “dietary intake of calcium phosophate and vitamin D” (Reviewer 2);
3) discuss in more detail “the potential effects and mechanisms of local mechano-response in bone on systemic components” (reviewer 2);
4) describe in more details “the effects of different types of training on systemic vitamin D and local FGF23 expression” (Reviewer 2);
5) provide additional methodological information on tissue preparation and mRNA extraction (Reviewer 2);
6) checking the text for typos (Reviewer 1);

Please also find below my comments which I would like you to consider:

1)the introduction is quite lengthy. Compatibly with reviewers’ needs, it should be limited to the essential information required to formulate the hypothesis driving the study and point out the pathway of interest;
2) the criteria used to determine the sample size should be indicated;
3) the statistical analysis section needs major attention. Did the authors check for normality distribution? It is true that normality test will unlikely detect non-Gaussian distribution with small sample sizes of 10 or fewer observations. However, since population distribution is normal for many variables in nature, unless normality is rejected, you should still use a parametric test that is more powerful than their equivalent non-parametric counterparts and can detect smaller differences. Also consider that, due to the central limit theorem, many parametric tests, including ANOVA, use the mean of the sample, so some non-normality can be tolerated;
4) the discussion should be more open to alternative explanations or, at least, provide tentative explanations for surprising or inconclusive results and, particularly, for discrepancies with previous studies (e.g. mRNA expression and serum FGF23 concentrations following exercise);
4) please expand the paragraph with weaknesses / limitations, including technical limitations (see for example the technical issues listed in the first and second subsections of “Results”) that may have somehow affected the results in the “Discussion” section;
5) the conclusions should be limited to those supported by the results. Speculation is welcomed but should be restricted to the discussion and clearly identified as such.

I hope that the referees' evaluation and my suggestions will be helpful should you decide resubmit a revised version of the manuscript.

Sincerely yours,

Stefano Menini

Reviewer 1 ·

Basic reporting

The authors described the effects of different training modalities on phosphate homeostasis in a rat model. They analysed four experimental groups: control rats; rats subjected to peak power (PT); animals subjected to endurance training (ET) and a group of rats subjected to a peak power followed by an endurance training (PET).
The paper could be interesting but there are many issues that should be clarified. Moreover, some typing error should be checked (for example line 96 and 337).

Experimental design

The experimental design is well described. However since FGF23 is a phosphaturic hormone, the measurement of phosphate in urine should be assessed. Moreover, in order to assess phosphate homeostasis and its correlation with PTH and FGF23, serum vitamin D levels should be measured.

Validity of the findings

The paper lacks of significant relevance since no differences were revealed between training-subjected rats and controls, except for PTH serum levels.

Additional comments

The authors submitted the paper entitled: “Effects of different training modalities on phosphate homeostasis and local vitamin D metabolism in rat bone”. Their experimental design is well described and the research question is meaningful. However some issue should be clarified as reported in previous sections.

Reviewer 2 ·

Basic reporting

The authors showed that neither FGF23 mRNA and protein expression levels in cortical bone nor FGF23 plasma concentrations differed between the groups. Additionally, they demonstrated that the mRNA expression levels of SOST, DMP1, PHEX and MEPE were lower after PT in comparison to ET and PET. Moreover, expression levels of CYP27b1 and VDR were also decreased after PT compared to ET. Author observed no differences in mRNA expression of CYO27b1, CYP24, VDR, NaPi-IIa and Na-Pi-IIc contransporters. The authors analyze serum levels of FGF-23, PTH, calcium and phosphate concentrations, but only serum PTH concentrations were higher after PT in comparison to controls.
1. In the ‘Introduction’ section line 97-100 – the authors should describe more precisely and widely the effects of different types of training on systemic vitamin D and local FGF23 expression.
2. Discussion - last paragraph: The authors note their key finding that the high intensity peak training may possible have caused changes in local phosphate regulation and vitamin D signaling within bone compared to endurance training. However, this finding deserves a bit more comment about the potential effects and mechanisms of local mechano-response in bone on systemic component.

Experimental design

1. The dietary intake of calcium phosophate and vitamin D were not described.

2. The authors should also measure the concentration of 1,25-di-hydroxyvitamin D3 [1,25(OH)2D3] in the serum of rats.

3. In the ‘Materials and methods’ section: Immunochemistry and Tissue preparation and mRNA extraction, the authors should precise which parts of the right tibiae (it was a proximal/ distal part of tibia?, it was the same region in each rat?) were taken under the investigation.

4. The number of animals is too small.

5. Authors should comment on the associations between examined parametres, if they are.

Validity of the findings

no comments

---

## Round 0.2 · accepted · Accept

Dear Dr. Bravenboer,

Our referees have now considered your paper and have recommended publication in “PeerJ”. We are pleased to accept your paper in its current form which will now be forwarded to the publisher for copy editing and typesetting.

I thank all reviewers for their effort in improving the manuscript and the authors for their cooperation throughout the review process

Yours sincerely,

Stefano Menini

# Reviewer 1 ·

Basic reporting

The authors described the effects of different training modalities on phosphate homeostasis in a rat model. They analysed four experimental groups: control rats; rats subjected to peak power (PT); animals subjected to endurance training (ET) and a group of rats subjected to a peak power followed by an endurance training (PET).
In the reviewed paper many issues have been clarified and typos have be checked.

Experimental design

The experimental design is well described. The authors added data that lacked in the previous version, for example serum vitamin D levels measurement, as suggested. Moreover they explained why phosphate in urine could not be evaluated.

Validity of the findings

In the reviewed version of the paper conclusion are limited to supporting results. Moreover the authors provided alternative explanations for inconclusive results and discrepancies, thus most of issues not well clarified in the previous version have been specified.

Additional comments

In the paper entitled: “Effects of different training modalities on phosphate homeostasis and local vitamin D metabolism in rat bone" experimental design is well described and the research question is meaningful. In the reviewed version the most of the issues has been clarified and discrepancies and limitations of the study have been specified.

Reviewer 2 ·

Basic reporting

In general, the authors did a good job of addressing the original review. However, the Authors should prepare the improved manuscript properly. In this file the Reviewer should receive clear text without deletions of the text, it impedes reading of the manuscript. All changes implemented in the text sholud be marked in red colour.

Experimental design

No comment

Validity of the findings

No comment

Additional comments

No comment